# Critically Ill Patients with Newly Diagnosed Anti-Neutrophil Cytoplasmic Antibody-Associated Vasculitis: Case Series and Literature Review

**DOI:** 10.3390/jcm13195688

**Published:** 2024-09-25

**Authors:** Kresimir Rukavina, Ozrenka Zlopasa, Ivana Vukovic Brinar, Feda Dzubur, Branimir Anic, Ana Vujaklija Brajkovic

**Affiliations:** 1Department of Internal Medicine, University Hospital Center Zagreb, Kispaticeva 12, 10000 Zagreb, Croatia; ozlopasa@gmail.com (O.Z.); ivbrinar@gmail.com (I.V.B.); banic@mef.hr (B.A.); avujaklija@gmail.com (A.V.B.); 2School of Medicine, University of Zagreb, Salata 3, 10000 Zagreb, Croatia; 3Department of Pulmonary Diseases, University Hospital Center Zagreb, Jordanovac 104, 10000 Zagreb, Croatia; fedja1104@gmail.com

**Keywords:** anti-neutrophil cytoplasmic antibody-associated vasculitis (ANCA-AAV), kidney failure, respiratory failure, diffuse alveolar hemorrhage, extracorporeal membrane oxygenation (ECMO), case report/series

## Abstract

ANCA-associated vasculitides (AAVs) are rare diseases with a prevalence of less than 200 cases per million persons and an incidence of less than 25 cases per million person-years. Their presenting features can vary from prodromal and nonspecific symptoms to dramatic organ-specific symptoms such as respiratory failure due to diffuse alveolar hemorrhage (DAH) and acute kidney injury (AKI). The latter two are hallmark features of pulmonary-renal syndrome, a potentially fatal condition that necessitates early recognition and treatment in intensive care units (ICUs) and rapid induction of immunosuppressive therapy. **Background and case summaries**: We described three patients with newly diagnosed AAV during the treatment of critical illness. All patients had DAH and two had AKI. The initial disease severity was extremely high in patients with myeloperoxidase (MPO)-AAV, reaching Sequential Organ Failure Assessment (SOFA) scores of 15 and 14 with predicted mortality ≥ 95.2%. Both patients needed mechanical ventilation, one additional venovenous extracorporeal membrane oxygenation (VV-ECMO), and renal replacement therapy. The patient with proteinase 3 (PR3)-AAV had a less severe disease, SOFA 3, requiring only modest oxygen supplementation and exhibiting only hematuria with normal renal function parameters. Immunosuppressive therapy was initiated during the ICU stay. The patient with the most severe clinical presentation died during the ICU stay because of sepsis, and the other two patients were discharged home. **Conclusions**: Patients with AAV presenting with pulmonary-renal syndrome necessitate various degrees of organ support. Nevertheless, these patients can be successfully treated in the early, critical stages of the disease and achieve remission.

## 1. Introduction

Respiratory insufficiency is one of the most common causes of intensive care unit (ICU) admission [1,2] with a variety of possible causes. Acute kidney injury (AKI) is considered a common occurrence in critically ill patients, with incidence reaching 60% [3]. When respiratory insufficiency is associated with AKI, the clinician should consider pulmonary-renal syndrome (PRS), which is defined by diffuse alveolar hemorrhage (DAH) and rapid progressive glomerulonephritis (RPGN). One of the three most common causes of PRS is anti-neutrophil cytoplasmic antibody (ANCA)-associated vasculitides [4]. ANCA-associated vasculitides (AAV) are rare diseases with a prevalence of less than 200 cases per million persons and an incidence of less than 25 cases per million person-years [5]. The following three types of small vessel vasculitides are associated with ANCAs: granulomatosis with polyangiitis (GPA), microscopic polyangiitis (MPA), and eosinophilic granulomatosis with polyangiitis (EGPA/formerly Churg–Strauss Syndrome) [6]. Both GPA and MPA commonly cause RPGN and/or DAH. Respiratory insufficiency due to DAH might be considered the most dramatic clinical manifestation of AAV. Patients necessitate different levels of respiratory support, from oxygen supplementation via nasal prongs to invasive mechanical ventilation and even extracorporeal membrane oxygenation (ECMO) to maintain gas exchange, thereby providing time for the administration of immunosuppressive therapy to control the inflammation [7,8,9]. The diagnosis of AAV is challenging and demanding because it must be fast and accurate as these conditions carry significant morbidity and mortality [10]. In a very short period, sometimes within hours, more common diseases, especially infections, should be ruled out and immunosuppressive treatment should be implemented. This is because, even despite advancements in the treatment of AAV, from cyclophosphamide to rituximab, the prognosis of patients with AAV is still unfavorable, reaching mortality up to 15% within the first year of the disease. The main causes of early death are infections and vasculitis manifestations, the latter up to 40%, and in most cases, due to DAH, as observed by Demiselle et al. [11]. Herein, we report three cases of AAV vasculitis with diffuse alveolar hemorrhage and renal dysfunction, which necessitated advanced organ support and treatment in the intensive care unit.

## 2. Case Presentations

### 2.1. Case No. 1

A 58-year-old Caucasian female patient was admitted to her local hospital because of hemoptysis lasting for several days and fatigue. The patient had no chronic disease and was not taking any chronic therapy. In the emergency department, the patient’s condition rapidly deteriorated. She developed severe respiratory insufficiency that necessitated orotracheal intubation and mechanical ventilation. A chest CT (Figure 1a) revealed bilateral pulmonary infiltrates with ground glass opacities and diffuse consolidation of the right lung. The initial laboratory and clinical findings are shown in Table 1. Because of the rapid development of respiratory failure, followed by bilateral pulmonary infiltrates, and a decrease in hemoglobin values, DAH was suspected. Empiric broad-spectrum antibiotic therapy with meropenem and vancomycin was initiated. The kidney function also deteriorated promptly. The differential diagnosis of pulmonary-renal syndrome was suspected, and the patient was transferred to the ICU of a tertiary care center. Bronchoscopy with bronchoalveolar lavage (Figure 1b) confirmed the diagnosis of DAH.

Mechanical ventilation for acute respiratory distress syndrome (ARDS) was introduced. However, the oxygenation of the patient was inadequate. The prone positioning did not improve oxygenation. Therefore, venovenous extracorporeal membrane oxygenation (VV ECMO) support commenced. The patient was hypotensive, and vasopressor therapy with noradrenaline was necessary. The initial Sequential Organ Failure Assessment (SOFA) score was 15 (predicted mortality ≥ 95.2%). The patient received a pulse dose of methylprednisolone (MP) (1 g IV for three consecutive days, followed by 2 mg/kg IV) and therapeutic plasma exchange. Myeloperoxidase (MPO)-ANCA were positive, and anti-glomerular basement membrane (anti-GBM) antibodies were negative, pointing toward a diagnosis of microscopic polyangiitis with a Birmingham Vasculitis Severity (BVAS) score of 27. Cyclophosphamide (CYC; 15 mg/kg IV) was introduced, followed by rituximab (RTX) (dose of 1 g IV) and an immunomodulatory course (2 g/kg) of intravenous immunoglobulins because of disease severity. The bleeding from the upper respiratory tract continuously complicated the treatment of the patient. In some periods, the ECMO was run without anticoagulation. However, the patient showed a certain respiratory improvement and was successfully weaned off ECMO after 3 weeks. Microbiology tests were repetitively positive despite antimicrobial therapy. *Acinetobacter baumanii*, *Pseudomonas aeruginosa*, and *Candida glabrata* were isolated from the bronchoalveolar aspirate. The patient was treated with colistin, cefepime, ampicillin with sulbactame, and anidulafungin. She was continuously dialysis-dependent without urine output. Liver necrosis was an additional complication. The patient died after 45 days in the ICU because of sepsis and multiorgan failure. During the ICU stay, the immunosuppressive therapy included the following: 2000 mg of RTX (1000 mg per dose, two weeks apart), three doses of CYC (15 mg /kg IV, i.e., 1000 mg IV per dose, two weeks apart), and glucocorticoids (pulse MP 1000 mg IV for 3 days, afterward 2 mg/kg IV, tapered to 1 mg/kg IV).

### 2.2. Case No. 2

A 64-year-old Caucasian male patient was admitted to the local hospital because of hematuria. The history included unexplained hematuria and chronic kidney disease (CKD) stage 3B (serum creatinine (sCr) 210 µmol/L, eGFR 30 mL/min/1.73 m^2^) diagnosed five months before indexed hospitalization. Chronic therapy included calcitriol, sevelamer, febuxostat, pantoprazole, and amlodipine. During the hospital stay, he developed a low-grade fever. Empiric antibiotic treatment with ceftriaxone was initiated. Anemia was corrected with a blood transfusion. A few days later, the patient became dyspnoeic and developed hemoptysis. The chest X-ray showed bilateral pulmonary infiltrates. Once again, he developed worsened anemia, necessitating a blood transfusion. In total, the patient received nine units of packed red blood cells. Levofloxacin was added to the therapy, but the C-reactive protein continued to increase, and the patient’s respiratory status deteriorated (Table 1). A chest CT (Figure 2a,b) showed massive bilateral interstitial infiltrates with ground glass opacities.

Because of the progression of respiratory insufficiency, oxygen supplementation via high-flow nasal cannula (HFNC) was initiated together with renal replacement therapy (RRT). A differential diagnosis of rapidly progressive respiratory insufficiency and renal failure pointed toward pulmonary-renal syndrome, and the patient was transferred to a tertiary hospital center. Upon admission to the ICU of the tertiary center, the patient was orotracheally intubated and mechanically ventilated, demanding significant respiratory support. The oxygenation goals were reached with prone positioning. Bronchoscopy with bronchoalveolar lavage confirmed the diagnosis of DAH. Because of hypotension, vasopressor therapy with noradrenaline (0.25 µg/kg/min) was necessary. RRT was continued, antibiotic therapy was modified (meropenem, linezolide, levofloxacine), and renal biopsy was performed. The initial SOFA score was 14 (predicted mortality ≥ 95.2%). Microbiology tests (blood culture, bronchoalveolar aspirate, urine culture) remained negative. A high dose of glucocorticoids (MP 500 mg IV) was introduced. The immunology results were positive for MPO-ANCA, antinuclear antibodies (ANAs), and antihistone antibodies. The values of C3 and C4 were normal. The pathohystological analysis showed moderate chronic changes in tubulointerstitium and glomeruli with superimposed glomerulonephritis with crescents and signs of acute tubular damage indicating a subacute phase of the disease, but still highly active. The conclusion of the pathohystological analysis pointed toward ANCA vasculitis. Summarizing the clinical course and kidney histology findings, the diagnosis of AAV, primarily MPA, was most plausible; therefore, it was concluded that ANA and antihistone antibodies were most likely false positive. Cyclophosphamide, 15 mg/kg IV, was introduced. Within several days of immunosuppressive therapy, the patient’s condition significantly improved. He was weaned from mechanical ventilation and RRT. A detailed history taken from the patient revealed several bouts of low-grade hemoptysis within the last few months, as well as the presence of hematuria and increased creatinine levels for several months. The patient had no elements indicative of other connective tissue diseases (apart from AAV). After two doses of CYC (1000 mg per dose), the patient showed significant clinical improvement. He was discharged from the ICU after 15 days and 8 days later from the hospital. Serum creatinine at discharge was 434 µmol/L, eGFR < 15 mL/min/1.73 m^2^. At the follow-up, oral prednisolone was tapered, and recently, the patient received the sixth and final dose of CYC in the induction cycle in the outpatient rheumatology clinic. In the ICU, his initial BVAS was 18 and decreased to 5 on the follow-up exam. RTX is planned for remission maintenance. Three months after hospital discharge, the renal function was somewhat improved (sCr is 229 µmol/L, eGFR 25 mL/min/1.73 m^2^) with persistent hematuria.

### 2.3. Case No. 3

A 46-year-old female Caucasian was admitted to a local hospital because of tachypnea, hemoptysis, and malaise. The patient’s prior medical history was positive for several bouts of epistaxis and bilateral otitis media. The initial chest X-ray showed bilateral pulmonary infiltrates, and a chest CT scan pointed toward either diffuse alveolar hemorrhage or cryptogenic organizing pneumonia (COP). The otolaryngologist described a vulnerable nasal mucosa with chronic atrophic changes. The patient was initially treated with levofloxacin. However, the hemoptysis progressed with a rapid decrease of oxygen saturation (SpO_2_ to 68% while breathing ambient air), and she became febrile (>38 °C) with an increase in inflammatory markers (Table 1). Oxygen supplementation via a non-rebreather mask (8–10 L/O_2_) and meropenem were introduced. A repeated chest CT showed progression of the pulmonary infiltrates now characterized as DAH. Considering clinical presentation, history, and signs of dysmorphic erythrocyturia, a differential diagnosis of a vasculitis-induced pulmonary-renal syndrome was made. The patient was transferred to the pulmonary ICU of a tertiary hospital center. The initial SOFA score was 3 (predicted mortality < 33.3%). Microbiology tests (blood culture, sputum, urine culture) remained negative. Proteinase 3 (PR3)-ANCA was positive, confirming the clinical diagnosis of granulomatosis with polyangiitis. A kidney biopsy was deemed unnecessary since the diagnosis of PR3-AAV was clear. The patient received high-dose glucocorticoids (MP 1000 mg per day for 3 consecutive days followed by 2 mg/kg IV) and RTX (375 mg/m^2^ IV weekly for four consecutive weeks), showing rapid clinical and laboratory improvement. The oxygen supplementation was weaned. The patient was discharged home after 14 days, and the final two induction doses of RTX were administered in the rheumatology outpatient clinic. Three months after hospital discharge, the patient was feeling well, without any clinical or laboratory signs of active vasculitis, and with a Birmingham Vasculitis Severity (BVAS) score of 0 (the initial BVAS was 20).

## 3. Discussion

In this case series, we described three patients with acute respiratory insufficiency and concurrent acute kidney disease presenting to local hospitals. All three patients had rapid clinical deterioration, primarily due to respiratory insufficiency caused by diffuse alveolar hemorrhage, which necessitated ICU admission and, in two cases, prompt invasive mechanical ventilation. Because of the accompanying derangement of renal function, pulmonary-renal syndrome was suspected. PRS was first described in 1919 by Goodpasture, and the term “Goodpasture’s syndrome” was adopted in 1958. In the late 1960s, the pathogenic role of anti-GBM antibodies was described in some cases. However, with time, the eponymous term “Goodpasture’s syndrome” was abandoned since it was discovered that a variety of conditions can lead to PRS and that most cases (about 70%) are caused by AAV [4,12,13]. As mentioned before, AAV includes three clinical syndromes; however, GPA and MPA most commonly affect both the respiratory tract and the kidneys, the latter being affected in up to 100% of cases [12,13]. Along with organ-specific symptoms, AAV patients often present with nonspecific symptoms such as fever, malaise, weight loss, myalgia, and arthralgia, and these prodromal symptoms can be present for weeks or even months without evidence of specific organ involvement [14]. The two types of ANCAs are anti-PR3 and anti-MPO. PR3-ANCAs are predominant antibodies in GPA (85%), and MPO-ANCAs in MPA (75–97%). Also, ANCAs can be found in other inflammatory conditions and infectious diseases, and they can even be drug-induced, necessitating rational application of the test in clinical practice [15]. Clinical manifestations, treatment response, relapse rates, and prognosis, according to growing evidence accumulated in the last few years, differ depending on the type of positive ANCA [13,15]. Patients who are PR3-ANCA-positive tend to have far more extrarenal manifestations than MPO-ANCA-positive patients [13], especially ear, nose, and throat involvement, which is often the first complaint among these patients. In a study published by Bantis et al., MPO-AAV often presented with advanced renal damage, and about 50% of patients required RRT upon presentation. Also, renal biopsies in MPO-AAV showed an increased frequency of chronic changes. On the other hand, according to the same study, PR3-AAV patients required initial RRT in less than 25% of cases. As for DAH, it was equally present in both groups [13] as one of the most serious complications of AAV and one of the strongest predictors of early mortality [16]. Patients with DAH due to immune causes, such as AAV, tend to be younger, have more severe presentations of the disease, and worse outcomes. In-hospital mortality of immune DAH is variable (12.5%) [17] and increases with the necessity of mechanical ventilation (77%) and renal replacement therapy (50%) [18]. In our case series, both MPA patients, consistent with the mentioned data, had a shorter course of nonspecific symptoms before PRS, and they had more severe renal involvement than the GPA patient as they required RRT. Also, the second patient had histological confirmation of both acute and chronic kidney changes; therefore, we could claim that the prior CKD was caused by subclinical AAV. As for DAH, it was equally the cause of rapid respiratory deterioration in all three cases, and the GPA patient had proof of nasal and sinus involvement and a history of recurrent middle ear infections for few years before developing PRS. As mentioned earlier, a kidney biopsy was performed only in one patient, and diffuse glomerulosclerotic changes were described with superimposed cresecentic glomerulonephritis and acute tubulointerstitial damage. Immunofluorescent microscopy also detected immune complexes bound to not sclerosed parts of glomeruli, and it is of note that according to the literature, such changes are occasionally seen and are associated with more severe disease [19,20]. The first patient received continuous anticoagulation because of the ECMO procedure, and biopsy was not possible; in the case of the third patient, clinical, imaging, and laboratory findings were sufficient to establish the diagnosis of AAV. Also, the patient had an excellent and prompt response to treatment, so the biopsy was declared to be an unnecessary risk without any additional benefits to the diagnosis or the patient. In all three cases, immunosuppressive treatment was initiated in the ICU setting, and the recent 2021 ACR/VF guidelines [19] and 2023 EULAR recommendations [15] were followed. However, the approach differed between patients, primarily because of the severity of the initial clinical presentation. All three patients initially received high-dose glucocorticoids (MP 500 to 1000 mg IV per dose), followed by dose reduction tapered individually according to the initial response to treatment. Apart from glucocorticoids, they were treated with either CYC, RTX, or both. MPO-ANCA patients were treated with CYC (15 mg/kg IV per biweekly dose) and in the case of the most severely ill patient requiring VV-ECMO because of prolonged DAH and profound respiratory failure, CYC was combined with RTX (1000 mg IV per dose, two weeks apart). Although there are still no sufficient data to vote for or against combining CYC and RTX [15], we decided to use both medications for remission induction because of the disease severity. This patient was also given an immunomodulatory course of intravenous immunoglobulin, according to the above-mentioned literature, to provide short-term control of the disease while waiting for the effect of the given induction remission therapy and because the patient had a significantly increased risk of infection. Also, it is important to mention that initially, with pulse glucocorticoids, she was treated with plasma exchange since there was a possibility of anti-GBM disease; however, plasma exchange was terminated upon detecting only positive MPO-ANCA. The patient’s renal function never improved but her respiratory function did; thus, the ICU team was able to de-escalate her from VV-ECMO to mechanical ventilation. It is noteworthy to mention that not many patients develop such pronounced respiratory failure demanding ECMO. Until 2021, there were 33 patients identified with AAV who were effectively treated with ECMO. The described patients were relatively young (average age was 32.4 ± 17.5 and 36.0 ± 16.1 years for males and females), and ECMO was initiated early, on the first day of ICU admission [7]. We can claim that ECMO in our patient helped to provide the necessary time for the application of immunosuppressive therapy but unfortunately, in the end, as a complication of both the disease and immunosuppression, she developed Gram-negative sepsis and multiorgan failure, from which she passed away. The other MPO-ANCA patient, although also in need of mechanical ventilation and RRT, responded far better to the applied immunosuppression (glucocorticoids and CYC). After finishing six cycles of CYC, he is in disease remission and is due for maintenance with RTX. Finally, the third patient, with positive PR3-ANCA, was treated initially with glucocorticoids and RTX dosed as in the RAVE trial (375 mg/m^2^ weekly for four consecutive weeks) [21]. The decision to choose RTX over CYC was made because the clinical presentation was not as severe as in the cases of the two MPO-ANCA patients and because PR3-ANCA are associated with higher relapse rates, for which RTX is recommended over CYC [15,21,22]. The patient promptly improved with treatment, never developed respiratory failure as severe as the other two patients, and had no renal involvement apart from massive hematuria, which soon became mild to moderate and is now absent according to the recent follow-up laboratory results. Lastly, it is important to note that none of the patients received avacopan. Although avacopan is recommended as a steroid-sparing agent and was shown to be a better aid in kidney recovery than glucocorticoids both for MPA and GPA patients [15,22], it is still unavailable in our country. Other medications, such as eculizumab [23,24] and tocilizumab [25], are mentioned in some case reports as a possible therapeutic option for patients with AAV. However, probably because of insufficient data and lack of clinical trials, neither is included in the recent treatment guidelines or the recommendations provided by ACR/VF [22] and EULAR [15] and, therefore, were not considered as treatment options for our patients. Furthermore, data have shown that AAV patients treated with C5 blockade (eculizumab) are at a higher risk of infection, especially with *Neisseria meningitidis*, despite the use of vaccination protocols and prophylactic antimicrobials. This is why the therapeutic targeting shifted to C5a, which, in the end, brought avacopan as one of the recommended therapeutic options [23].

## 4. Conclusions

We have presented three patients (patient summaries available in the Appendix A) who were admitted to intensive care units because of pulmonary-renal syndrome and were diagnosed with ANCA-associated vasculitis. The patients with MPO-AAV had a more rapid and severe deterioration of both respiratory and renal function, and the PR3-AAV patient had a less dramatic but nevertheless severe clinical presentation. Both patients with MPO-AAV had extremely severe presentations of the disease, which was seen in a high SOFA score that in critically ill patients is associated with mortality of more than 92%. They required RRT and mechanical ventilation, one of them even being escalated to VV-ECMO, which helped to bridge the early stage of the disease until immunosuppression gained some effect. Despite significant disease severity and bad prognosis, one patient recovered almost completely. The PR3-ANCA-positive patient had a prior medical history of recurrent ear and upper respiratory tract inflammatory episodes, which concurs with the common disease evolution. Also, both the pulmonary and kidney involvement were less severe and not as progressive when compared with the MPO-ANCA-positive patients. Through our case series, we have shown that clinicians need to be aware that AAV patients can present with severe clinical manifestations such as PRS and can rapidly deteriorate necessitating admission to the ICU. They also need to be aware that a multidisciplinary team consisting of an ICU specialist, a pulmonologist, a nephrologist, and a rheumatologist can quickly diagnose and treat patients with AAV.

## Figures and Tables

**Figure 1 jcm-13-05688-f001:**
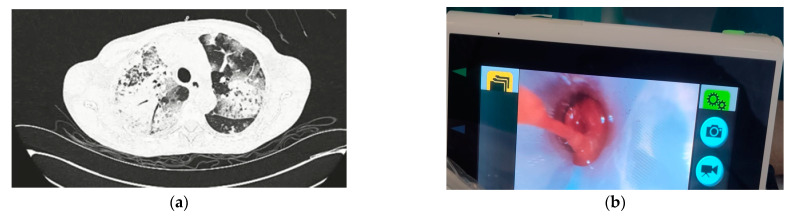
(**a**) Chest CT showing bilateral and diffuse pulmonary infiltrates and consolidations described as changes due to diffuse alveolar hemorrhage. (**b**) Videobronchoscopy image showing lung hemorrhage.

**Figure 2 jcm-13-05688-f002:**
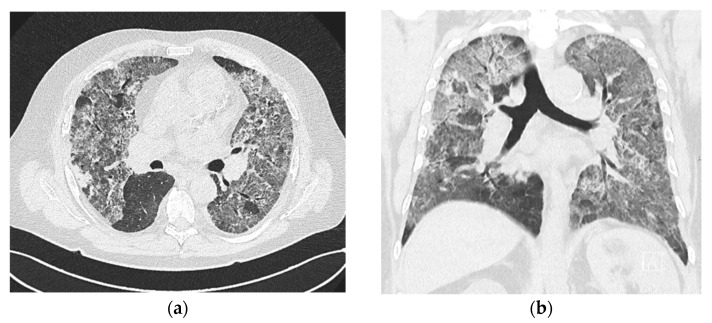
(**a**) Transverse chest CT scan showing bilateral lung changes due to diffuse alveolar hemorrhage. (**b**) Chest CT scan of the same patient, coronal plane image.

**Table 1 jcm-13-05688-t001:** Initial relevant laboratory results from the emergency department and upon admission to the ICU.

		Patient No. 1		Patient No. 2		Patient No. 3	
	Reference Range	ED	ICU	ED	ICU	ED	ICU
RBCs (×10^12^/L)	3.86–5.08	2.52	2.70	2.26	2.93	3.66	3.8
Hemoglobin, g/L	119–157	76	78	66	84	116	116
WBCs (×10^9^/L)	3.4–9.7	7.4	8	4.44	6.1	6.2	6.2
Leukocyte Differential Count (×10^9^/L)							
ANC	2.06–6.49	6.1	7.43	3.69	5.53	4.07	5.82
Lymphocytes	1.19–3.35	0.94	0.4	0.27	0.28	1.39	0.23
Monocytes	0.12–0.84	0.23	0.17	0.36	0.23	0.54	0.12
Basophils	0.0–0.6	0.01	0.0	0.03	0.03	0.03	0.01
Eosinophils	0.0–0.43	0.04	0.0	0.09	0.03	0.17	0.01
Platelets (×10^9^/L)	158–424	286	222	169	166	199	142
PT	>0.7	0.87	0.98	0.96	1.01	1.29	1
Fibrinogen, g/L	1.8–4.1	6.31	4.5	6.81	>7.0	5.69	>7.0
LDH, U/L	<241	N/A	257	458	544	N/A	155
Urea, mmol/L	2.8–8.3	52.9	31.5	31.2	36.9	4.7	7.7
Creatinine, µmol/L	60–104	1406	848	660	690	71	66
Urinalysis							
E	neg.	+++	Anuria	+++	+++	+++	+++
LE	neg.	++	Anuria	+/−	neg.	+	neg.
Prot.	neg.	++	Anuria	+	+	+	+/−
CRP, mg/L	<5	124.1	96.7	237.7	269.3	7.1	269.3
Procalcitonin, ug/L	<0.25	3	2.7	N/A	5.71	0.16	N/A

Abbreviations: ED: emergency department; ICU: intensive care unit; RBC: red blood count; WBC: white blood count; ANC: absolute neutrophil count; LDH: lactate dehydrogenase; E: eryhtrocyte; LE: leukocyte esterase; Prot.: protein; CRP: C-reactive protein; PT: prothrombin time.; Urinalysis (E, LE, Prot) was performed by dipstick method and the grading system is as follows: “negative”, trace (“+/−“), positive/detectable (“+”), moderate (“++”), high grade (“+++”).

## Data Availability

All data are available in the hospital information system.

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
