# Peer review of "Critically Ill Patients with Newly Diagnosed Anti-Neutrophil Cytoplasmic Antibody-Associated Vasculitis: Case Series and Literature Review"

_jcm, 2024, doi:10.3390/jcm13195688_

Round 1

Reviewer 1 Report

Comments and Suggestions for Authors

The study "Critically Ill Patients with Newly Diagnosed ANCA-Associated Vasculitis: Case Series and Literature Review" presented three cases with ANCA-associated vasculitis presenting with severe forms of pulmonary-renal syndrome. The study used a standard approach to presenting the cases and the authors explained all the important findings in the Discussion section in a very meticulous way. Additionally, the Discussion section evaluates the clinical and laboratory findings of the presented patients in detail and pays attention to all the important aspects of their meaning and potential roots.

Conclusions correspond with the results obtained and emphasize some of the suggestions for future actions in order to increase the probability for successful recovery of patients with such a severe manifestations.

There are no major suggestions for improvement of the presentation of patients in the manuscript.

The authors should check if all the abbreviations are explained on their first use in both the abstract and the manuscript (some of them are missing, like for MPO and ECMO).

However, due to the fact that this area has already been discussed in a literature, the authors could try to further emphasize (in the Discussion and Conclusion sections, as well as in the Abstract) the severity of manifestations as a novel description of possible clinical image in ANCA-positive patients with pulmonary-renal syndrome.

Overall, the quality and the importance of the findings of this article are very good and, other than mentioned, no further corrections are necessary.

Author Response

Dear Editor and reviewers,

Thank You very much for your comments.  We have accepted comments and changed the text accordingly improving the overall quality of the manuscript. Detailed responses to the comments follow. We hope that you will find the manuscript adequate for publication.

Reviewer 1.

Comment: The authors should check if all the abbreviations are explained on their first use in both the abstract and the manuscript (some of them are missing, like for MPO and ECMO).

Answer: Done as suggested.

Comment: However, due to the fact that this area has already been discussed in a literature, the authors could try to further emphasize (in the Discussion and Conclusion sections, as well as in the Abstract) the severity of manifestations as a novel description of possible clinical image in ANCA-positive patients with pulmonary-renal syndrome.

Answer: The comment was taken into consideration, and we have tried to further emphasize the importance of early recognition and treatment of PRS in AAV. Changes have been made in lines 17 and 18 (p. 1), and lines 305 to 307 (p. 8).

The manuscript, with the changes, is in the attachement.

Reviewer 2 Report

Comments and Suggestions for Authors

Dear Authors!

Thank you for the opportunity to reveiw your manuscrip.

ANCA-vasculitis is a rare but life-theratening disease with predominant lung and kidney involvement with poor outcomes and high mortality.

The manuscript is a cased-based review of three critically ill patients with ANCA vasculitis with lung and kindey involvement. Authors provide the sufficient introduction about  actuality of this problem and described three cases in details and provided the literature review of the same condition

The conclusion supported the results

During the revision I have several suggestions

1) Please provide the additional table with treatment or add the treatment information in the existed table

2) Due to case-based review standard will be interesting to created the table with liteture data about similar disease. it will be useful for the readers to see the short version of described trials or case series.

3) Also provide information about alternative treatment, such as eculizumab in the cases of acute kibdey injury and about new treatment for ANCA-vasculitis - avopacan and using of other treatment - tocilizumab etc. 

Author Response

Dear Editor and reviewers,

Thank You very much for your comments.  We have accepted comments and changed the text accordingly improving the overall quality of the manuscript. Detailed responses to the comments follow. We hope that you will find the manuscript adequate for publication.

Reviewer 2.

Comment: Please provide the additional table with treatment or add the treatment information in the existed table

Answer: The course of the treatment for every patient have been summarized in a table already available as a supplementary material (Table 1 -3). We added the detailed treatment protocol in the table.

Comment: Due to case-based review standard will be interesting to create the table with literature data about similar disease. it will be useful for the readers to see the short version of described trials or case series.

Answer:

Dear reviewer, could you please elaborate your comment in more detail, specifically what diseases do you have in mind?

Regarding the mention of similar diseases (as a probable cause of PRS), we primarily focused on pulmonary-renal syndrome in AAV, since our patients have all been diagnosed with PRS due to new-onset AAV, and since according to our literature review AAV is one of the most common causes of PRS.  As for the trials part in the comment, would you like us to include a short summary of clinical trials in AAV (for instance MAINRITSAN trials and RAVE trial)?

Comment: Also provide information about alternative treatment, such as eculizumab in the cases of acute kidney injury and about new treatment for ANCA-vasculitis - avopacan and using of other treatment - tocilizumab etc.

Answer: The comment was taken into consideration and an additional paragraph was added at the end of the “Discussion” (p. 7, lines 281-289): “Other medications, such as eculizumab [23,24] and tocilizumab [25] are mentioned in some case reports as a possible therapeutic option for patients with AAV. However, probably due to insufficient data and a lack of clinical trials, neither is included in the recent treatment guidelines and recommendations published by ACR/VF [22] and EULAR [15], and therefore were not considered as treatment options for our patients. Furthermore, data has shown that AAV patients treated with C5 blockade (eculizumab) are at a higher risk of infection, especially with Neisseria meningitidis, despite the use of vaccination protocols and prophylactic antimicrobials, which is why the therapeutic targeting shifted to C5a which in the end brought avacopan as one of the recommended therapeutic options [23].”

Also, three additional references were included: 23, 24 and 25.

The manuscript, with the changes, is attached here as well as on the main upload site along with the revised supplementary material.

Round 2

Reviewer 2 Report

Comments and Suggestions for Authors

Dear Authors!

Thank you for the revised version

I have no additional comments